# Identification of Deregulated Proteins in Mutated *BRCA1*/*2* Breast and Ovarian Cancers for Vectorized Biologics

**DOI:** 10.3390/cancers17132208

**Published:** 2025-07-01

**Authors:** Adrián Sanvicente, Cristina Nieto-Jiménez, Esther Cabañas Morafraile, Cristina Díaz-Tejeiro, Vanesa García Barberán, Pedro Pérez Segura, Győrffy Balázs, Alberto Ocaña

**Affiliations:** 1Experimental Therapeutics in Cancer Unit, Medical Oncology Department, Hospital Clínico San Carlos (HCSC), Instituto de Investigación Sanitaria San Carlos (IdISSC), 28040 Madrid, Spain; adriansanvicenteg@gmail.com (A.S.); cnietoj@salud.madrid.org (C.N.-J.); ecmorafraile@gmail.com (E.C.M.); cristinadiaztejeiro@gmail.com (C.D.-T.); 2Facultad de Ciencias Químicas, Universidad Complutense de Madrid, 28040 Madrid, Spain; 3Center for Biological Research Margarita Salas (CIB-CSIC), Spanish National Research Council, 28049 Madrid, Spain; 4Molecular Oncology Laboratory, Hospital Clínico San Carlos (HCSC), Instituto de Investigación Sanitaria San Carlos (IdISSC), 28040 Madrid, Spain; vanesa.garciabar@salud.madrid.org; 5Medical Oncology Department, Hospital Clínico San Carlos (HCSC), Instituto de Investigación Sanitaria San Carlos (IdISSC), 28040 Madrid, Spain; pedro.perez@salud.madrid.org; 6Department of Bioinformatics, Semmelweis University, Tűzoltó U. 7-9, 1094 Budapest, Hungary; gyorffy.balazs@yahoo.com; 7Research Centre for Natural Sciences, Hungarian Research Network, Magyar Tudosok Korutja 2, 1117 Budapest, Hungary; 8Department of Biophysics, Medical School, University of Pecs, 7624 Pecs, Hungary; 9START, Hospital Fundación Jiménez Díaz, Cátedra START-INTHEOS-CEU, Universidad San Pablo CEU, 28040 Madrid, Spain

**Keywords:** BRCA1, BRCA2, ADCC, breast cancer, ovarian cancer, immune population

## Abstract

Breast and ovarian cancers with mutations in the BRCA1 and BRCA2 genes are especially sensitive to PARP inhibitors. These drugs block DNA repair mechanisms, causing cancer cell death. However, PARP inhibitors have demonstrated certain toxicity to healthy cells, limiting their safe use. To improve the precision and safety of these treatments, we sought to identify specific proteins on the surface of cancer cells that carry BRCA mutations. These proteins could serve as “flags” for targeted therapies, delivering treatment directly to cancer cells while sparing healthy tissues. We analyzed public datasets comparing gene activity in cancers with and without BRCA mutations. We identified several surface proteins that were more abundant in BRCA-mutated cancers. Some of these are linked to patient outcomes and the immune system, suggesting a potential role in cancer progression and immune response. By identifying these proteins, we can develop new therapies that use antibodies to deliver drugs, like PARP inhibitors, directly to cancer cells, reducing side effects and improving the effectiveness of current treatments for BRCA-mutated cancers.

## 1. Introduction

Targeting membrane proteins in cancer cells with antibodies has shown clinical activity in many different clinical scenarios. For instance, when targeting *HER2* overexpressed breast cancers with trastuzumab or colorectal and head and neck cancers with cetuximab, an improvement in patient survival has been observed [1]. This strategy induces two types of non-clinical pharmacological activities: one is the inhibition of the oncogenic effect of the target itself, and the second is the activation of the immune system [2,3,4,5].

Notably, this strategy is not without side effects, mainly if the target is expressed in non-tumoral cells, inducing on-target non-tumoral toxicity [6]. To avoid this limitation, the selection of proteins that are at higher levels in the membrane of tumor cells or the extracellular matrix is mandatory. A good example is the targeting of *HER2* in breast cancer, where overexpression of the HER2 protein due to gene amplification is observed [7,8].

A further step is the use of antibodies to selectively deliver toxic agents to tumor cells. Antibody-drug conjugates (ADCs) allow the vectorization of chemical entities for chemotherapy [9]. Several examples have reached the clinical setting, such as those acting against the membrane protein TROP2, a cellular receptor involved in calcium signaling and linked to oncogenesis, in triple-negative breast cancer [10] or *ErbB2* in *HER2*-positive breast cancer [8,11]. Following the path for the vectorization of chemotherapy, one would wonder why other chemical entities that have demonstrated activity but harbor a narrow therapeutic index should not be vectorized [1,12]. Targets for vectorized biologics may include surface and extracellular proteins, as well as intracellular components, when peptide-based delivery systems are used.

Germline mutations in *BRCA1* and *BRCA2* predispose patients to around 40–70% of breast and ovarian cancers, making preventive strategies crucial for these patients [13]. In addition, cancers harboring these mutations have impaired DNA repair mechanisms, as these genes code for proteins that participate in DNA double-strand break (DSB) repair functions. This alteration makes these cancers susceptible to agents like platinum compounds. Cells with *BRCA1*/*2* mutations use the homologous recombination (HR) pathway to repair DNA [14], in which PARP proteins play a central role. Cancers with these mutations are more vulnerable to PARP inhibitors (PARPi), a process known as synthetic lethality. Synthetic lethality refers to a situation in which the simultaneous alteration of two genes or proteins causes cell death, in our particular case, BRCA1/2 and PARP [15,16,17]. Although this strategy is scientifically relevant, its translation into a clinical setting has limitations, particularly in the development of combinations. For instance, the association of PARPi with DNA-damaging chemotherapies displays a high toxicity profile, limiting its general implementation in several indications [18]. A general strategy to improve this problem would be to vectorize compounds with antibodies to act directly on tumor cells or their microenvironment, avoiding the side effects produced on non-tumor cells, thereby increasing their therapeutic index.

In our work, we aimed to identify proteins specifically upregulated in mutated *BRCA1* and *BRCA2* breast and ovarian cancers that could be used as targets for the development of novel therapies.

## 2. Materials and Methods

### 2.1. Identification of BRCA1/BRCA2 Mutations in Breast and Ovarian Cancer Patients

The initial data for this study were obtained from TCGA (https://www.cancer.gov/tcga (accessed on 17 November 2024)) to investigate gene alterations when *BRCA1* or *BRCA2* is mutated in patients with ovarian or breast cancer. The number of cases analyzed varied between breast and ovarian cancers. In the case of breast cancer, expression data from 979 patients were used, of which 23 were BRCA1 mutants and 24 were BRCA2 mutants. In the case of ovarian cancer, expression data from 272 patients were used, of which 13 were BRCA1 mutants and 12 were BRCA2 mutants. Most of the ovarian cancer cases were stage III or stage IV (76% and 17%, respectively), with few stage II cases (6%). About 75% of breast cancer cases and 90% of ovarian cancer cases were White, and 18% and 7%, respectively, were African American. Statistical significance was assessed using the Mann–Whitney U test.

Surface protein expression was assessed using the Human Surface Atlas (http://wlab.ethz.ch/surfaceome/, accessed on 21 November 2024) [19].

### 2.2. Analysis of Molecular Function of the Selected Genes

To assess the functional annotation of our genes, we used the public online platform “EnrichR” (https://maayanlab.cloud/Enrichr/, accessed on 21 November 2024) [18,19,20]. Biological functions were obtained from the Gene Ontology of those genes that were regulated and present on the cell surface for each dataset.

### 2.3. Outcome and Prognosis Analysis

The relationship between the expression of certain genes and clinical outcome in patients with ovarian or breast cancer was addressed using the KM Plotter Online tool [20] (https:/kmplot.com/analysis/, accessed on 23 November 2024). This database allows the assessment of the Overall Survival (OS) of patients with upregulation of selected genes. A total of 1220 samples for ovarian cancer (OS and Grade 3 + 4) and 1879 for breast cancer (OS) were analyzed. A False Discovery Rate (FDR) ≤10% and statistical significance were selected. This web tool uses Cox proportional hazards regression to assess statistical significance.

### 2.4. Immune Cell Infiltration and Gene Expression Correlation

To investigate the association between gene expression and immune infiltration, the TIMER2.0 web tool [21,22] (http://timer.cistrome.org/, accessed on 24 November 2024) was used. TIMER2.0 uses Spearman’s correlation to associate gene expression with immune populations.

### 2.5. Statistical Analysis

Statistical significance was assessed using the Mann–Whitney U test applied to the TCGA gene expression data.

Enrichment analysis was performed using Enrichr, which provides its own statistical framework.

Survival analysis, including hazard ratios and *p*-values, was conducted using the KM Plotter tool, which employs Cox proportional hazards regression.

Correlation analysis was performed using the TIMER platform (version TIMER2.0), which applies Spearman’s correlation.

### 2.6. Graphical Design

Histograms, bar charts, heat maps, dot plots, and volcano plots were generated using GraphPad Prism software (version 8.0.1) (GraphPad Software, San Diego, CA, USA).

## 3. Results

### 3.1. Flow Chart for the Selection and Identification of Genes

We compared the transcriptomic profiles of breast and ovarian cancers harboring mutations in *BRCA1* and *BRCA2* to identify differential transcriptomic patterns of expression. To do so, we interrogated public datasets to analyze genes upregulated or downregulated when these mutations were present, compared to those with wild-type *BRCA1*/*2* cancers. We calculated the mean expression in both groups and used the Mann–Whitney U test to assess the statistical significance. Using an arbitrary cut-off (fold change (FC) ≥ 2) and a statistically significant difference (*p* < 0.01), as described in the graphical abstract, we identified 11 upregulated and 44 downregulated transcripts in *BRCA1*-mut breast cancer, and 10 upregulated and 57 downregulated transcripts in *BRCA2*-mut breast cancer. In ovarian cancer, 79 genes were upregulated and 123 were downregulated in *BRCA1-mut* cancers, and five were upregulated and seven were downregulated in *BRCA2-mut* cancers. The workflow followed in this study is depicted in the graphical abstract.

### 3.2. Selection of the Relevant Upregulated Genes in Breast and Ovarian Cancer

The next step was to identify the genes present in the cell membrane according to the Human Surfaceome Atlas [19]. We found that 69 and 10 of our upregulated genes in *BRCA1*-mut breast and ovarian cancer, respectively, were expressed on the cell surface. For *BRCA2-mut*, only two and nine genes had surface expression and were upregulated in breast and ovarian cancers, respectively (Figure 1A).

Some genes were particularly highly expressed. For instance, in *BRCA1-mut* ovarian cancer, *KTR14*, *KRT16*, *CXCL9*, and *CFD* were highly upregulated (more than 4-fold change), and in *BRCA2-mut* ovarian cancers, *TSPAN7* and *CDKN2C* were clearly upregulated (more than 2-fold change) (Figure 1B). In *BRCA1*-mut breast cancer, *KANK4*, *MFGE8*, *LINC00346*, and *A100A1* were upregulated (more than 3-fold change), and in *BRCA2-mut* breast cancers, *MAGEA4* was highly expressed (more than 4-fold change) (Figure 1B).

### 3.3. Gene-Set Enrichment Analysis of Upregulated Genes

In *BRCA1*-mutated ovarian cancers, the main functions identified were related to the immune system, such as MHC assembly, CD4 and CD25 regulation, and regulation of the interferon gamma pathway, among others (Figure 2A). In *BRCA2*-mut ovarian tumors, functions included regulation of phosphorylation, vesicle transport, and regulation of signaling (Figure 2B). In contrast to ovarian cancers, most functions in BRCA1-mut breast cancer were related to cell damage repair and angiogenesis (Figure 2C). In the case of *BRCA2*-mut breast cancer (Figure 2D), most pathways were also related to the immune system, including cytokine production, inflammasome regulation, interferon response, and T-cell migration, among others.

### 3.4. Outcome Analysis and Differential Expression of Upregulated Surfaceome Genes

Next, we evaluated whether the expression of any of the identified genes was associated with patient survival. To do so, we matched the expression of the genes with the outcome. Figure 3 shows dot plots that describe the risk of death in relation to the outcome (Hazard Ratio, HR), without considering the false discovery rate (FDR), for each of the evaluated genes in *BRCA1* and *BRCA2*, breast, and ovarian cancer.

In ovarian cancer, *BRCA1*-mut cancer genes such as AIF1, *CD8A*, and *BST1* showed detrimental prognosis, while in *BRCA2*-mut ovarian cancer, *SEC61A2* and *IGFBP3* were associated with shorter survival (Figure 3A). In breast *BRCA1*-mut, *CD109, AHNAK2*, and *MFGE8* were selected as transcripts associated with detrimental outcomes. In contrast, for *BRCA2*-mut breast cancer (Figure 3B), *B3GNT7*, *CTSV*, and *GSDMC* were the most significant.

These data suggest that these genes could potentially be druggable targets for the development of antibodies due to their differential expression in these tumors and their association with a detrimental prognosis. In this context, we evaluated the expression of these genes in other solid cancers compared to that in normal tissues. To this end, we interrogated TCGA data, as described in the Materials and Methods section. Some of these genes were significantly more expressed in tumors than in normal tissues. *IGFBP3* was highly expressed in several cancers, including ACC, BLCA, ESCA, GBM, HNSC, LUAD, LUSC, PAAD, STAD, and *AIF1* in DLBC, PAAD, SKCM, and THYM. Others showed expression in a limited number of indications, such as *CD8A* in THYM, *AHNAK2* in KIRP, LUSC, and PAAD, *MFGE8* in DLBC, HNSC, PAAD, SKCM, THYM, and *CTSV* in CESC, HNSC, and THYM.

### 3.5. Identification of Outcome-Related Surfaceome Biomarkers

We then evaluated the genes that predicted outcomes with the condition of an FDR < 10%, aiming to identify prognostic markers with higher accuracy. Using this strict approach, we selected two genes associated with a detrimental prognosis: *B3GNT7* and *CTSV* in *BRCA2*-mut breast cancers, and three with a favorable prognosis: *CD6*, *CXCL9*, and *CXCL13* in *BRCA1*-mut ovarian cancers (Figure 4A).

The combinations of the identified genes were also explored. For genes that predicted a favorable prognosis (CD6, CXCL9, and CXCL13), no improvement was observed (Figure 4B). In contrast, the combination of *B3GNT7* and *CTSV* predicted detrimental survival (HR 2.02 (1.54–2.64), *p* < 0.001), compared with the evaluation of single genes (Figure 4C).

Finally, given the immunological roles of *CD6*, *CXCL9*, and *CXCL13*, we explored their potential associations with immune populations. In ovarian cancer, we observed a correlation with effector T and dendritic cell infiltration (Figure 5).

## 4. Discussion

In the present article, we identified upregulated genes expressed in the surfaceome of breast and ovarian cancers with mutations in the *BRCA1* and *BRCA2* genes. From our analysis, only a minority of genes complied with the criteria: 69 and 10 genes in *BRCA1* and *BRCA2* mutated breast cancers, respectively, and 10 and 9 genes in *BRCA1* and *BRCA2* mutated ovarian cancers, respectively.

If a more restricted cut-off was selected, the number of genes was clearly reduced. Only *KTR14*, *KRT16*, *CXCL9*, and *CFD* in *BRCA1* Ovarian Cancer and *TSPAN7* and *CDKN2C* in *BRCA2* ovarian cancers were highly upregulated, and *KANK4*, *MFGE8*, *LINC00346*, and *SA100A1* in *BRCA1* mutated breast cancer, and *MAGEA4* in *BRCA2* breast cancers. Interestingly, no genes were commonly shared between the subtypes.

This finding is relevant as it suggests that the evaluated mutations modify biological functions that translate into modifications of the surfaceome or extracellular matrix of the cell in a differential manner. Therefore, some of the identified genes may also be upregulated in other tumor types compared to normal tissues. Consistent with this, we observed that functions were not shared among subtypes, and only in the *BRCA1* breast cancer subtype, immune biological functions were redundant among genes. The most relevant functions included *MHC class II-related functions and chemokine and cytokine activity*. In the other subtypes, the small number of identified genes represented a variety of biological functions. This suggests that although these genes are implicated in DNA repair mechanisms, the biological transcriptomic phenotype differs between them, suggesting the importance of co-occurring genomic alterations, including polymorphisms, and not only a single mutation, to define a characteristic oncogenic phenotype [23].

When evaluating surfaceome genes, we identified some that were upregulated and associated with detrimental outcomes. These include *AIF1*, *CD8A*, and *BST1* in ovarian cancer *BRCA1*-mutant cancers, and SEC61A2 and *IGFBP3* in *BRCA2*-mutant ovarian cancer. *CD109, AHNAK2*, and *MFGE8* in breast *BRCA1*-mut cancers, and *B3GNT7*, *CTSV*, and *GSDMC* in *BRCA2*-mut breast tumors. We then explored the expression of these genes in tumors compared with that in normal tissues. Notably, some genes were clearly upregulated in cancer samples, as was the case for *IGFBP3*. This gene belongs to the insulin-like growth factor binding protein (*IGFBP*) family and encodes a protein that forms a ternary complex with the insulin-like growth factor acid-labile subunit (IGFALS) and either insulin-like growth factor (*IGF*) I or II [24].

Notably, some genes were associated with a particular detrimental or favorable prognosis with a low false discovery rate, suggesting that these genes could be used as biomarkers of prognosis. For instance, *B3GNT7* and *CTSV* predicted detrimental prognosis in *BRCA2*-mut breast cancers, and *CD6*, *CXCL9*, and *CXCL13* predicted favorable outcomes in *BRCA1*-mut ovarian cancers. Indeed, the combined signature of *B3GNT7* and *CTSV* predicted worse outcomes than either gene alone.

Finally, as the last three genes were associated with MHC class II-related functions and chemokine and cytokine activity, we explored their association with immune populations. *CD6*, *CXCL9*, and *CXCL13* were correlated with effector T and dendritic cell infiltration. In addition, CXCL13 is present with neutrophils in the tumor. CD6 is a gene involved in cell adhesion and cell-cell contacts, promoting T-cell activation and proliferation and regulating T-cell response, contributing to the triggering of TCR/CD3 cascades [25]. *CD6* has been considered a potential therapeutic target [26]. *CXCL9* and *CXCL13* are part of a chemokine superfamily of proteins involved in immune recruitment, immunoregulatory, and inflammatory processes [27,28]. Mutations in BRCA1/2 cause inefficiency in homologous recombination repair, leading to genomic instability and the accumulation of damaged DNA. This condition activates innate immune pathways, such as the cGAS–STING pathway, promoting type I interferon responses. Interferon cascade activation induces the production of chemokines, such as CXCL9, which promotes the recruitment of CD8^+^ T cells to the tumor microenvironment and stimulates the expression of CXCL13, intensifying immune infiltration and improving the response to checkpoint inhibitor treatments [28].

Several aspects should be considered globally. First, mutations in *BRCA1*/*2* do not significantly modify the surfaceome of the cell in a characteristic manner. Secondly, these genomic alterations do not affect similar biological pathways and functions. Thirdly, these mutations do not significantly influence the immune microenvironment, except in ovarian *BRCA1*-mutant cancers where the upregulation of chemokines and antigen-presenting cell components is observed.

This study has certain limitations and should be interpreted as hypothesis-generating, given its reliance on publicly available genomic datasets and therefore the lack of functional validation. The scarcity of comparable transcriptomic studies on BRCA1/2-mutated tumors limits the possibility of direct benchmarking. Nonetheless, by evaluating both extracellular and intracellular components—each representing potential targets for vectorized therapeutic strategies—we sought to provide a broad perspective on targetable alterations. Further validation, including protein-level analyses of patient-derived samples, is essential to confirm and extend these findings.

Although our analysis focused on gene expression and survival associations, future studies could expand on these findings by exploring post-translational modification (PTM) signatures associated with the identified genes, as such modifications may offer deeper insights into their functional roles in tumor progression.

In summary, we identified candidate genes expressed on the membrane surface that could be used for therapeutic intervention or prognostic stratification, as they were clearly upregulated and differentially expressed between cancers and normal tissues in a wide range of solid cancers.

## 5. Conclusions

We identified membrane, humbly, and extracellular matrix-associated genes specifically deregulated in BRCA1- and BRCA2-mutated breast and ovarian cancers through genomic mapping of public datasets. Several of these genes, including B3GNT7, CTSV, CD6, CXCL9, and CXCL13, were prognostically significant and associated with immune cell infiltration. These findings highlight potential novel targets for therapeutic delivery.

## Figures and Tables

**Figure 1 cancers-17-02208-f001:**
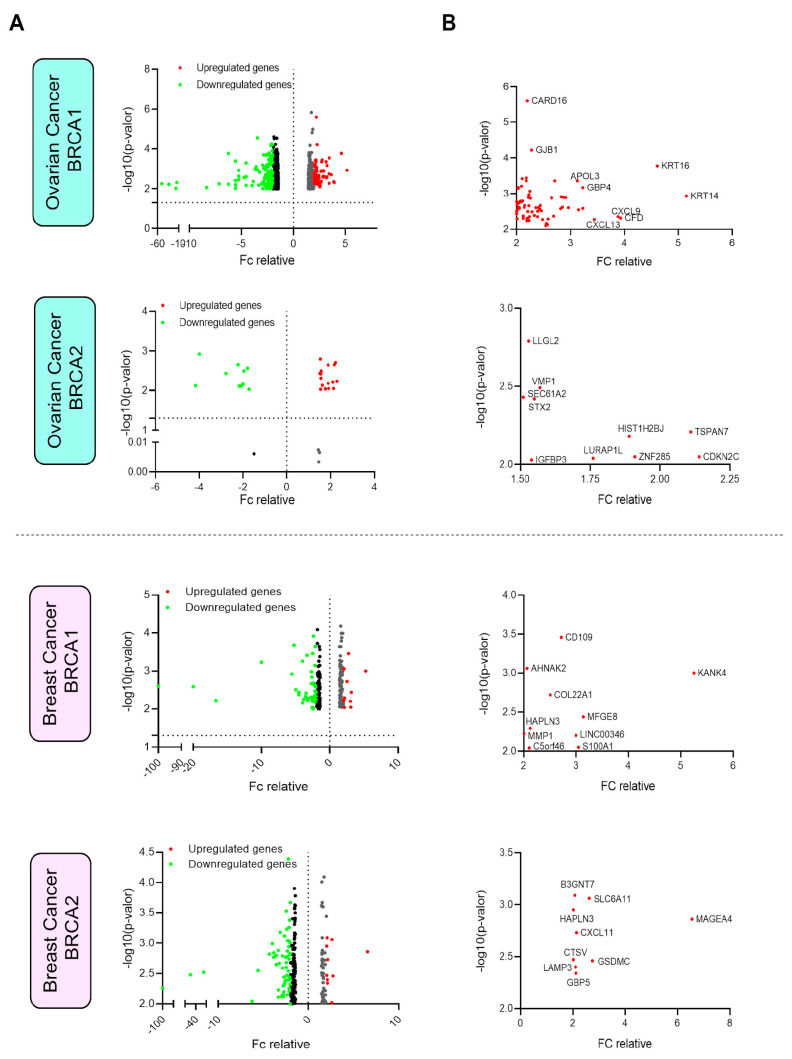
Analysis of differential gene expression in ovarian and breast cancers mutated in *BRCA1* and *BRCA2*. (**A**) Vulcan plots showing the expression and fold change (FC) of each gene in both ovarian and breast cancers with mutated *BRCA1*/*2*. (**B**) Details of the genes selected with an FC greater than 2 and a *p*-value less than 0.01 (assessed by the Mann–Whitney U test).

**Figure 2 cancers-17-02208-f002:**
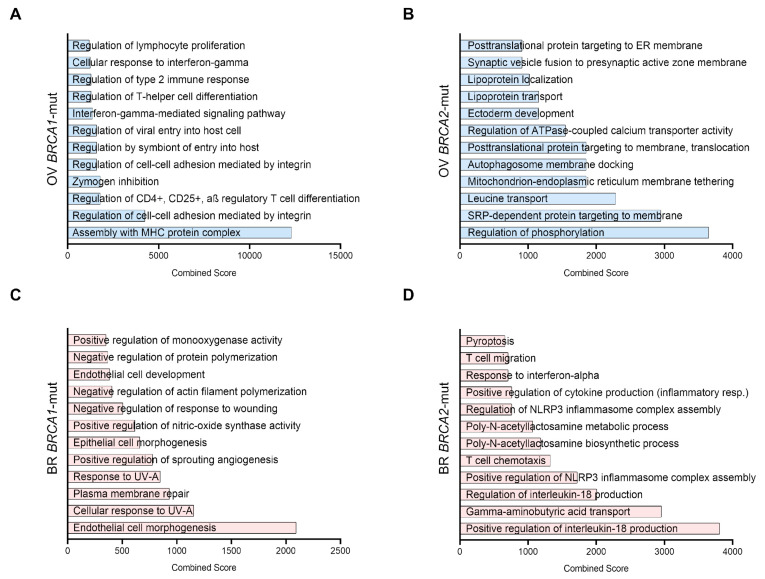
Functional enrichment of the selected genes. Most significant biological functions related to each set of selected genes (those that are upregulated and with surface expression) organized by Combined Score (According to EnrichR, the Combined Score is computed by taking the log (*p*-value) from the Fisher exact test and multiplying by the z-score of the derivation from the expected rank). (**A**) Ovarian Cancer *BRCA1* mutated genes (**B**), Ovarian Cancer *BRCA2* mutated genes (**C**), Breast Cancer *BRCA1* mutated genes (**D**), Breast Cancer *BRCA2* mutated genes.

**Figure 3 cancers-17-02208-f003:**
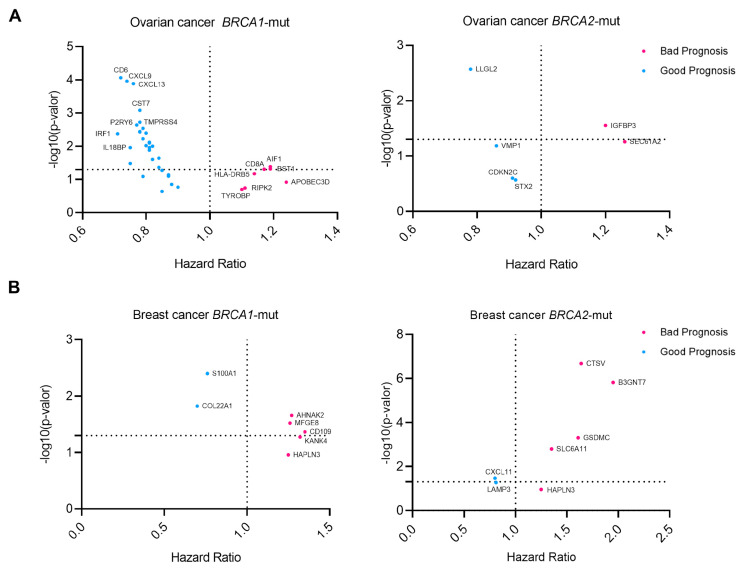
Prognostic data for selected genes showing information on the Hazard Ratio and gene expression of each gene (survival data from KM Plotter). A Hazard Ratio above 1 indicates a detrimental effect of the expression on survival, while a hazard ratio below 1 indicates a good prognosis. Cox proportional hazards regression analysis was performed for each gene separately (**A**), Ovarian Cancer *BRCA1*/*2* mutated genes. (**B**) Breast Cancer *BRCA1*/*2* mutated genes.

**Figure 4 cancers-17-02208-f004:**
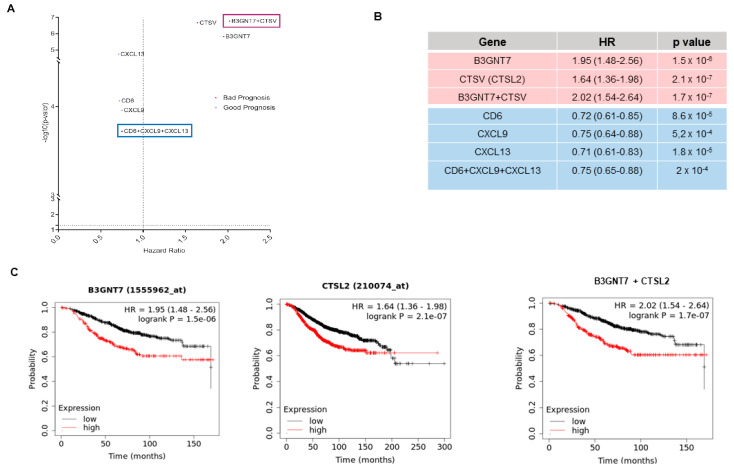
(**A**) Prognostic data for each of the most relevant genes in our datasets compared to the prognosis obtained for the signatures between them. (**B**) Details of the hazard ratios (with 95% confidence intervals in brackets) and the significance of the data shown in Figure 4. a. (**C**) Survival data as KM plots of the most relevant detrimental prognostic genes (*B3GNT7*, *CTSL2*, and their combined signature).

**Figure 5 cancers-17-02208-f005:**
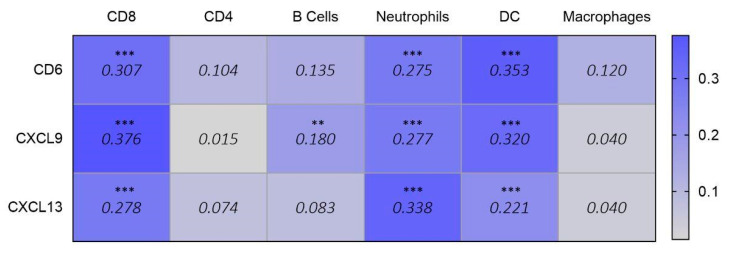
Correlation data (purity-adjusted Spearman’s rho) showing the association between the expression of selected signature genes (CD6, CXCL9, and CXCL13) and infiltrating immune populations in ovarian cancer (data obtained from TIMER 2.0). Values below 0.150 (shown in grey) are considered not significant. Values equal to or above 0.150 (shown in blue) are statistically significant (*p* < 0.01 **, *p* < 0.001 ***) and indicate a positive correlation (Spearman’s rho > 0) between gene expression and immune cell infiltration. For significant values, higher Rho values reflect stronger correlations.

## Data Availability

All data generated or analyzed during this study are included and appropriately cited in this published article.

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
