# Peer review of "Identification of Deregulated Proteins in Mutated BRCA1/2 Breast and Ovarian Cancers for Vectorized Biologics"

_cancers, 2025, doi:10.3390/cancers17132208_

Round 1

Reviewer 1 Report

Comments and Suggestions for Authors

The authors analyzed gene expression profiles in breast or ovarian cancer samples of patients who were carriers of germline mutations in BRCA1 or BRCA2 genes in order to identify membrane proteins which were specifically upregulated in BRCA+ cancers. The analysis was done on 1220 ovarian cancer samples and 1879 breast cancer samples downloaded from accessible databases. The No of samples was big enough for statistical analysis. They analyzed the expression of genes which was changed at least two-fold (FC>2) compared with wild-type samples. They identified two genes B36NT7 and CTSV connected with unfavorable prognosis in BRCA2- mut breast cancers and three genes CD6, CXCL9 and CXCL13 connected with favorable prognosis in BRCA1 – mut ovarian cancers. The comparison with clinical outcomes allowed them to find differentially expressed genes which could be potentially targets for anticancer therapy or prognostic evaluation. The advantage of the analysis of gene expression profiling is the complex analysis of expressed genes. The authors draw modest conclusions in respect to affected signaling pathways. The manuscript is interesting.

Author Response

Comments and Suggestions for Authors

The authors analyzed gene expression profiles in breast or ovarian cancer samples of patients who were carriers of germline mutations in BRCA1 or BRCA2 genes in order to identify membrane proteins which were specifically upregulated in BRCA+ cancers. The analysis was done on 1220 ovarian cancer samples and 1879 breast cancer samples downloaded from accessible databases. The No of samples was big enough for statistical analysis. They analyzed the expression of genes which was changed at least two-fold (FC>2) compared with wild-type samples. They identified two genes B36NT7 and CTSV connected with unfavorable prognosis in BRCA2- mut breast cancers and three genes CD6, CXCL9 and CXCL13 connected with favorable prognosis in BRCA1 – mut ovarian cancers. The comparison with clinical outcomes allowed them to find differentially expressed genes which could be potentially targets for anticancer therapy or prognostic evaluation. The advantage of the analysis of gene expression profiling is the complex analysis of expressed genes. The authors draw modest conclusions in respect to affected signaling pathways. The manuscript is interesting. 

Response: We acknowledge this observation and have expanded the discussion on this point in the revised manuscript. Lines 302-322. Thank you for your comments on our work.

Reviewer 2 Report

Comments and Suggestions for Authors

This is an interesting study. However, in the Introduction, some general data related to prevention of breast cancer in BRCA1/2 carriers are necessary. As an example, the National Comprehensive Cancer Network (NCCN) recommends that BRCA1/2 carriers be offered risk-reducing bilateral mastectomy. 

Author Response

Comments and Suggestions for Authors

This is an interesting study. However, in the Introduction, some general data related to prevention of breast cancer in BRCA1/2 carriers are necessary. As an example, the National Comprehensive Cancer Network (NCCN) recommends that BRCA1/2 carriers be offered risk-reducing bilateral mastectomy.

Response: We agree with this point. Preventive strategies are crucial for individuals carrying BRCA1/2 mutations, and current guidelines, such as those from the National Comprehensive Cancer Network (NCCN), include recommendations for risk-reducing interventions like bilateral mastectomy to lower breast cancer risk. We have added a brief commentary about this in the introduction. Thank you.

Reviewer 3 Report

Comments and Suggestions for Authors

General Comment: The manuscript presents an interesting and potentially valuable study that could contribute to the understanding of BRCA-mutated cancers and the identification of membrane protein targets. However, the manuscript in its current form has significant weaknesses in the areas of formatting and methodology that should be addressed before it can be considered for publication in Cancers.

Formatting Issues. The manuscript does not comply with several of the journal’s formatting requirements:

  • The Plain Language Summary is missing, which is mandatory for Cancers.

  • The Abstract exceeds the recommended length and should be shortened in accordance with MDPI guidelines.

  • References are not formatted according to MDPI style, both in-text and in the reference list.

  • The Conclusions section is missing, which is a required structural element of the journal.

These issues should ideally have been addressed during the technical check prior to peer review. The authors are encouraged to thoroughly revise the manuscript to ensure full compliance with the journal’s formatting standards.

Methodological Concerns.

  1. Introduction:
    The Introduction includes excessive background detail that could be condensed. A more concise and focused summary of the research context would enhance readability and scientific focus.

  2. Study Aim:
    The stated aim of the study is:
    "In our work, we aim to identify membrane proteins specifically upregulated in mutated BRCA1 and BRCA2 breast and ovarian cancers, with the main goal to develop antibodies that could be used to vectorize agents acting on DNA damage repair mechanisms like PARPi."
    However, the manuscript does not report the development of antibodies. Therefore, the aim should be reformulated to reflect the actual scope and findings of the study, such as identification of candidate membrane proteins as potential antibody targets.

  3. Statistical Analysis:
    The description of the statistical methods lacks clarity and consistency:

  • The authors mention the use of GraphPad Prism for data visualization but do not specify the statistical tests used in many cases.

  • P-values are reported without indicating the corresponding tests, except for one mention of Fisher’s exact test.

  • The manuscript refers to twofold or threefold differences—clarification is needed on how these thresholds were defined or statistically evaluated.

  • Hazard ratios are mentioned, but the methods used to obtain them (e.g., Cox proportional hazards regression) are not explained.

  • Figure 6 presents correlation analysis results, but the type of correlation test (e.g., Pearson, Spearman) is not specified. Additionally, statistical significance (p-values) and strength of associations (weak, moderate, strong) should be clearly stated or annotated using asterisks or other standard conventions.

  • Figure 5 displays hazard ratios with values in brackets (presumably 95% confidence intervals), but this should be explicitly stated. Also, some p-values are reported with excessive precision (e.g., p = 2.57E-20). It is more appropriate to report such values as p < 0.001.

A revised and detailed Statistical Analysis section is essential for transparency and reproducibility.

  1. Discussion Section:
    The discussion could benefit from more head-to-head comparisons with relevant prior studies, which would help contextualize the findings.
    The limitations of the study should be expanded to include discussion of factors such as the lack of functional validation, sample size constraints, potential selection bias, and generalizability.

Author Response

Comments and Suggestions for Authors

General Comment: The manuscript presents an interesting and potentially valuable study that could contribute to the understanding of BRCA-mutated cancers and the identification of membrane protein targets. However, the manuscript in its current form has significant weaknesses in the areas of formatting and methodology that should be addressed before it can be considered for publication in Cancers.

Formatting Issues. The manuscript does not comply with several of the journal’s formatting requirements:

The Plain Language Summary is missing, which is mandatory for Cancers.

Response: A plain language summary has been added in accordance with the instructions for authors. Thank you for your comment.

The Abstract exceeds the recommended length and should be shortened in accordance with MDPI guidelines.

Response: This problem has been fixed in accordance with MDPI guidelines. Thank you for your comment.

References are not formatted according to MDPI style, both in-text and in the reference list.

Response: References have been modified to MDPI style as suggested. Thank you for your comment.

The Conclusions section is missing, which is a required structural element of the journal.

Response: We have included a conclusion section summarizing the most relevant findings.

These issues should ideally have been addressed during the technical check prior to peer review. The authors are encouraged to thoroughly revise the manuscript to ensure full compliance with the journal’s formatting standards.

Methodological Concerns.

Introduction:

The Introduction includes excessive background detail that could be condensed. A more concise and focused summary of the research context would enhance readability and scientific focus.

Response: We have reviewed the introduction section and we have reduced some of the consent to provide a more comprehensive and condensed version.

Study Aim:

The stated aim of the study is:

"In our work, we aim to identify membrane proteins specifically upregulated in mutated BRCA1 and BRCA2 breast and ovarian cancers, with the main goal to develop antibodies that could be used to vectorize agents acting on DNA damage repair mechanisms like PARPi."

However, the manuscript does not report the development of antibodies. Therefore, the aim should be reformulated to reflect the actual scope and findings of the study, such as identification of candidate membrane proteins as potential antibody targets.

Response: We agree with this comment. Accordingly, we have reduced and condensed the Introduction section to provide a broader overview and to better highlight the study objectives. In line with this, we have also revised the title and the main goal of the manuscript to reflect a more general but accurate perspective: “Identification of Deregulated Proteins in BRCA1/2-Mutated Breast and Ovarian Cancers as Targets for Vectorized Biologics”.

Statistical Analysis:

The description of the statistical methods lacks clarity and consistency:

The authors mention the use of GraphPad Prism for data visualization but do not specify the statistical tests used in many cases. P-values are reported without indicating the corresponding tests, except for one mention of Fisher’s exact test.

Response: We appreciate the reviewer’s observation. We would like to clarify that GraphPad Prism was used solely for data visualization and not for performing any statistical analyses. The statistical tests reported throughout the manuscript were conducted using the respective web tools employed in each analysis, as outlined below:

  • Figure 1: Statistical significance was assessed using the Mann–Whitney U test applied to TCGA gene expression data.

  • Figure 2: Enrichment analysis was performed using EnrichR, which provides its own statistical framework.

  • Figures 3 and 4: Survival analysis, including hazard ratios and p-values, was conducted using the KM Plotter tool, which employs Cox proportional hazards regression.

  • Figure 5: Correlation analysis was performed using the TIMER platform, which applies Spearman’s correlation.

In order  to ensure that the source and type of each statistical test are clearly stated, we have added line 174 “(assessed by the Mann–Whitney U test.“, line 202 “Cox proportional hazards regression analysis was made for each gene separately” and line 241 “ Correlation data (purity-adjusted spearman's rho)”. Furthermore, we have added a similar explanation in the corresponding sections of Material & Methods.

The manuscript refers to twofold or threefold differences—clarification is needed on how these thresholds were defined or statistically evaluated.

Response: We thank the reviewer for highlighting the need for clarification regarding the use of twofold or threefold differences. In this manuscript, we compared the transcriptomic profiles of tumor samples from BRCA1/2-mutated patients versus non-mutated (wild-type) patients. For each gene, we calculated the mean expression in both groups and used the Mann–Whitney U test to assess statistical significance. Only genes showing a statistically significant difference in expression (p < 0.05) were considered for further analysis and representation. The terms “twofold” or “threefold” refer to the magnitude of change in average gene expression levels between groups and were used descriptively to highlight biologically relevant differences, not as predefined statistical thresholds.

We have revised the manuscript and added a sentence in line 147-8 to ensure it is clearly stated.

Hazard ratios are mentioned, but the methods used to obtain them (e.g., Cox proportional hazards regression) are not explained.

Response: We appreciate the reviewer’s comment regarding the methods used to obtain the hazard ratios. In our manuscript, we have focused on reporting the survival analysis results generated using the web-based tool, rather than detailing its internal methodology. However, we would like to clarify that the statistical method used—Cox proportional hazards regression—is indeed thoroughly described in the original publication of the tool we utilized (Győrffy et al., Comput Struct Biotechnol J., 2021; https://doi.org/10.1016/j.csbj.2021.07.014). As stated in that paper:

“Cox proportional hazards regression analysis was made for each gene separately. In this, each possible cutoff value was examined between the lower and upper quartiles, and False-Discovery Rate using the Benjamini-Hochberg method was computed to correct for multiple hypothesis testing. The survival analysis was performed for relapse-free survival (RFS)...”

Given that the tool's statistical basis is publicly available and validated, and our analysis strictly relies on its output, we aimed to present the biological and clinical implications of the results rather than reiterate the already published methodological framework. Nevertheless, if the reviewer deems it necessary, we are happy to include a brief summary of the underlying statistical approach and reference the original source accordingly.

Figure 6 presents correlation analysis results, but the type of correlation test (e.g., Pearson, Spearman) is not specified. Additionally, statistical significance (p-values) and strength of associations (weak, moderate, strong) should be clearly stated or annotated using asterisks or other standard conventions.

Response: Figure 5 (previously called Figure 6) presents data obtained from the online tool TIMER2.0. This tool uses Spearman's correlation to define correlations between immune infiltrate and the expression of specific genes. This information can be found in the text in the corresponding section of Materials & Methods (Immune cell infiltration and gene expression correlation). Furthermore, the figure has been modified in accordance with the reviewer's recommendations. Thank you for your comment.

Figure 5 displays hazard ratios with values in brackets (presumably 95% confidence intervals), but this should be explicitly stated. Also, some p-values are reported with excessive precision (e.g., p = 2.57E-20). It is more appropriate to report such values as p < 0.001.

Response: Thank you for your helpful comment. We confirm that the values in brackets following the hazard ratios in Figure 4 (previous Figure 5) represent 95% confidence intervals, and we have updated the figure legend in line 235 to explicitly state this. Additionally, we have revised the p-values throughout the manuscript to follow standard reporting conventions, expressing extremely small values as p < 0.001 (line 241).

Discussion Section:

The discussion could benefit from more head-to-head comparisons with relevant prior studies, which would help contextualize the findings.

The limitations of the study should be expanded to include discussion of factors such as the lack of functional validation, sample size constraints, potential selection bias, and generalizability. -

Response: We agree with the reviewer on the value of a more direct comparison with existing studies. However, to our knowledge, studies specifically evaluating the transcriptomic landscape of tumors harboring these mutations are currently lacking. Regarding the study’s limitations, we have included a dedicated paragraph in the Discussion section providing a detailed explanation of all potential confounding factors: “This study has certain limitations and should be interpreted as hypothesis-generating, given its reliance on publicly available genomic datasets. The scarcity of comparable transcriptomic studies in BRCA1/2-mutated tumors limits the possibility of direct benchmarking. Nonetheless, by evaluating both extracellular and intracellular components—each representing potential targets for vectorized therapeutic strategies—we sought to provide a broad perspective on targetable alterations. Further validation, including protein-level analyses in patient-derived samples, will be essential to confirm and extend these finding”

Reviewer 4 Report

Comments and Suggestions for Authors

This study is a bioinformatics exercise to identify surface genes between mutated and wildtype BRCA1 and BRCA2 ovarian and breast cancer. The text is well written, it is easy to read and understand. The authors correlated with the survival of the patients, but other clinicopathological characteristics of the series are not described. The authors are not using own data, and this can be seen as a weak point. However, this doesn't mean the results are worth publishing.

(1) Regarding trastuzumab and cetuximab, please describe the mechanism of action.

(2) Line 64.Regarding HER2. Could you please describe how HER2 is evaluated in breast cancer? If FISH or DISH is 3Black and 3Red (CEP17) the ratio is 1 and it is not considered amplification. However, there is copy number gain, so this is also "pathological gain"?

(3) Line 69. What is the function of trop2 protein?

(4) Section 2.3. Does KM plotter allows to select only mutated cases?

(5) Please add version of graphpad software.

(6) Could you please describe the clinicopathological characteristics of the series of section 3.1? How many cases were analyzed?

(7) In Figure 1, what is the meaning of the "exclamation symbol"? What is the color for mutation and non-mutated (whildtype)?

(8) Section 3.3. Please describe how GSEA was performed. Or is ti the EnrichR software?

(9)  Are the genes shown in figure 4 and section 3.4 surface-only?

(10) What markers are being used to identify cd8, cd4, bcell , etc. in Figure 6?

(11) How are the final identified genes (filtered by all the steps; i.e. the most relevant) related to the mutational status? They were identified by differential gene expression, but, how BRCA and CXCL9 relate?

(12) What is the role of the more relevant genes in other types of cancer?

Author Response

Comments and Suggestions for Authors

This study is a bioinformatics exercise to identify surface genes between mutated and wildtype BRCA1 and BRCA2 ovarian and breast cancer. The text is well written, it is easy to read and understand. The authors correlated with the survival of the patients, but other clinicopathological characteristics of the series are not described. The authors are not using own data, and this can be seen as a weak point. However, this doesn't mean the results are worth publishing.

(1) Regarding trastuzumab and cetuximab, please describe the mechanism of action. Response: We have described the mechanism when we discuss the dual effect, against an oncogene and activating the immune system: “This strategy induces two types of non-clinical pharmacology activity, one leaded by the inhibition of the oncogenic effect of the target itself, and the second one by activating the immune system”.

(2) Line 64. Regarding HER2. Could you please describe how HER2 is evaluated in breast cancer? If FISH or DISH is 3Black and 3Red (CEP17) the ratio is 1 and it is not considered amplification. However, there is copy number gain, so this is also "pathological gain"? -

Response: We appreciate this interesting insight. As far as we know, HER2 in breast cancer is evaluated using immunohistochemistry (IHC) to assess the level of HER2 protein expression on tumor cell membranes. Cases with an IHC score above 2 undergo additional testing by in situ hybridization (ISH) to determine HER2 gene amplification. In the context of cancer, copy number gain can lead to moderate overexpression of the gene which may be functionally relevant but not detectable with FISH. Moreover, these increases  may lead to partial resistance to targeted therapies and could justify both the inclusion of the patients in certain clinical trials or the precise treatment with new combined therapies.

The presence of increased HER2 copy number without classical amplification in breast cancer implies a sustained but moderate activation of HER2 related pathways and defines an emerging subgroup candidate for new targeted therapies, especially next-generation ADCs.

Therefore, even though the score does not reveal amplification, if there is an increase in the number of copies, the gain should be considered a pathological and clinically relevant feature.

We can add this information to the main text if deemed necessary and relevant to improve understanding of the manuscript.

(3) Line 69. What is the function of trop2 protein?

Response: TROP2 is a cell surface receptor involved in calcium signaling. It is encoded by the gene TACSTD2 (tumor associated calcium signal transducer 2) and its overexpression has been reported in various solid tumors, including breast cancer. When upregulated, this protein participates in oncogenic processes such as tumor proliferation, resistance to therapy and invasion capability. Targeted therapy against Trop2 is an emerging promise to treat several indications leading to the development of new drugs and the approval of an anti-Trop2 ADC (sacituzumab govitecan) to treat metastatic TNBC. Brief information about TROP2 has been added to the text in line 81.

Liu, Xinlin et al. “Advances in Trop2-targeted therapy: Novel agents and opportunities beyond breast cancer.” Pharmacology & therapeutics vol. 239 (2022): 108296. doi:10.1016/j.pharmthera.2022.108296

(4) Section 2.3. Does KM plotter allows to select only mutated cases?

Response: Thank you for your question. Unfortunately, KM Plotter does not currently allow filtering based on BRCA1/2 mutational status. The only mutational filter available within the tool is for TP53 status.

(5) Please add version of graphpad software.

Response: GraphPad version used has been added in the Material & Methods section (Graphical design and statistical analysis, Line 126). Thank you for your comment.

(6) Could you please describe the clinicopathological characteristics of the series of section 3.1? How many cases were analyzed?

Response: The number of cases analyzed varies between breast and ovarian cancer. In the case of breast cancer, expression data from 979 patients is used, of which 23 are BRCA1 mutants and 24 are BRCA2 mutants. In the case of ovarian cancer, expression data from 272 patients is used, of which 13 are BRCA1 mutants and 12 are BRCA2 mutants. The number of cases analyzed varies between breast and ovarian cancer. In the case of breast cancer, expression data from 979 patients is used, of which 23 are BRCA1 mutants and 24 are BRCA2 mutants. In the case of ovarian cancer, expression data from 272 patients is used, of which 13 are BRCA1 mutants and 12 are BRCA2 mutants. Most of the ovarian cancer cases had a stage III or stage IV disease (76% and 17%, respectively), with few stage II cases (6%). About 75% of breast cancer cases and 90% of ovarian cancer cases were White, and 18% and 7%, respectively, were African American. This information has been added to the text in the corresponding section of Materials and Methods (Identification of BRCA1/BRCA2 mutations in Breast and Ovarian Cancer Patients).

(7) In Figure 1, what is the meaning of the "exclamation symbol"? What is the color for mutation and non-mutated (wild type)?

Response: The exclamation mark is simply a way of visually representing the presence of the mutation in the BRCA1 or BRCA2 genes. As for the colors, the international colors for breast and ovarian cancer are used to represent in an easily understandable way which genes correspond to each set of patients. The graphical abstract shows only the altered genes, whether with higher or lower expression, in mutated BRCA cancer compared to wild-type BRCA, as these are the ones that will be worked with. Therefore, there is no representation or color associated with the wild type since it does not appear in the figure. Thank you for your comment, we hope this explanation helps to clarify our ideas.

(8) Section 3.3. Please describe how GSEA was performed. Or is it the EnrichR software?

Response: Our analysis of functional annotation is entirely done using EnrichR software as described in the Material & Methods section of the manuscript (Line 107 - Analysis of molecular function of the selected genes). Thank you for your comment.

(9)  Are the genes shown in figure 4 and section 3.4 surface-only?

Response: Thank you for your observation. The genes presented in Figure 4 and Section 3.4 are, indeed, not only exclusively surface proteins. In line with this, we have provided in the current version a more general and accurate overview to identify targets that could be used for different formats to vectorize biologics.

(10) What markers are being used to identify cd8, cd4, bcell , etc. in Figure 6?

Response: Data displayed in Figure 6 (actual Figure 5) has been obtained from TIMER2.0 database. The bibliography related to the software does not contemplate any information related to markers used to define these populations. However, these are widely known and well described populations; and data from TIMER2.0 relies on solid databases.

(11) How are the final identified genes (filtered by all the steps; i.e. the most relevant) related to the mutational status? They were identified by differential gene expression, but, how BRCA and CXCL9 relate?

Response: Thank you for your helpful comment. BRCA mutations (BRCA1 and BRCA2) impair homologous recombination repair, leading to increased genomic instability, accumulation of DNA damage, and production of cytosolic DNA fragments. This cytosolic DNA can activate the cGAS-STING pathway, resulting in type I interferon signaling and increased expression of immune-related chemokines, including CXCL9. CXCL9 is a chemokine that recruits effector T cells (particularly CD8+ T cells) into the tumor microenvironment. Therefore, BRCA-mutated tumors may show increased CXCL9 expression as part of an innate immune response to DNA damage, contributing to a more inflamed, potentially immunogenic tumor microenvironment.

(12) What is the role of the more relevant genes in other types of cancer?

Response:

B3GNT7: Involved in regulation of cell surface glycosylation. Alterations in the levels of this protein are described in several cancer types. In Colon cancer, its downregulation lead to enhanced invasion capability through augmented sLea/sLex antigens. In Breast cancer, its overexpression is linked to poor prognosis.

CTSV: Encode cathepsin V, a cysteine protease associated with oncogenesis. Its deregulation is associated with cell cycle deregulation, TME modulation (affecting immune cell infiltration), tumor growth and metastasis in several cancers including lung, breast and colorectal cancer among others.

CD6: Key gene for lymphocyte differentiation and activity. Present on T cells and other immune subsets as B cells or NK cells. Involved in the recognition of cancer related antigens as CD166 or CD318 and identified as objective for targeted therapy in T-cell lymphomas.

CXCL9: T cell attracting chemokine associated with a dual role in cancer modulating anti-tumor activity. Its role is not well understood. As an immune chemokine has a suppressive role through recruitment of effector T cells and, in some cancers, it is related with a better response to ICIs treatment. However, it has been described a role of CXCL9 in promoting tumor cell migration and metastasis in several cancers. Its role is highly context-dependent.

CXCL13: Similar to CXCL9, CXCL13 is involved in B cell recruitment and formation of germinal centers but also in tumor growth and metastasis. CXCL13 effects are influenced by cancer type, TME and signaling pathways cross talking. In breast cancer its overexpression is related to lymph node metastasis while in ovarian cancer it could serve as a predictive marker for enhanced immunotherapy efficacy.

We think it might be excessive background information taking into account the actual aim of this work. However, if needed, this information could be added to the main text.

Reviewer 5 Report

Comments and Suggestions for Authors

The manuscript by Adrián et al. examined candidate genes expressed on the membrane surface that may serve as targets for therapeutic intervention in cancer.

I have few comments for polishing the manuscript.

  1. The authors have found some potential upregulated genes associated with cancer. Did they identify any potential PTM signatures from the publicly available datasets associated with the genes.
  2. Did the authors find any common gene signatures associated with both the cancers?
  3. The role of immune cell infiltration is poorly discussed and not well depicted in the study. The authors should depict that in a clearer manner.
  4. The main goal of this study is to develop antibodies that could be used to vectorize agents acting on DNA damage repair mechanisms like PARPi. The authors did not explain this in their study in a clearer manner. The rationale is not clearly defined.

Author Response

Comments and Suggestions for Authors

The manuscript by Adrián et al. examined candidate genes expressed on the membrane surface that may serve as targets for therapeutic intervention in cancer.

I have few comments for polishing the manuscript.

The authors have found some potential upregulated genes associated with cancer. Did they identify any potential PTM signatures from the publicly available datasets associated with the genes.

Response: We appreciate the reviewer’s insightful question. In this study, our analysis focused primarily on differential gene expression and survival associations. We did not specifically investigate PTM signatures related to the deregulated genes in mutated BRC1/2. While some public datasets may contain PTM-related information, such as phosphorylation or acetylation data, these were beyond the scope of our current work. However, we agree that exploring PTM signatures could provide valuable functional insights and represents an important direction for future research. We have added a note acknowledging this in the discussion section (lines 325-329).

Did the authors find any common gene signatures associated with both the cancers?

Response: Indeed, this is something we analyzed when conducting the study, but we did not find any common deregulated gene signatures between ovarian cancer and breast cancer. It is explained in the discussion section. Thank you for your comment.

The role of immune cell infiltration is poorly discussed and not well depicted in the study. The authors should depict that in a clearer manner.

Response: Thank you for your helpful comment. We have provided a more detailed explanation adding an additional paragraph: “Mutations in BRCA1/2 cause inefficiency in homologous recombination repair, leading to genomic instability and accumulation of damaged DNA. This condition activates innate immune pathways as the cGAS–STING pathway, promoting type I interferon responses. The interferon cascade activation induces the production of chemokines such as CXCL9, which promote the recruitment of CD8⁺ T cells to the tumor microenvironment, and can stimulate the expression of CXCL13 intensifying immune infiltration and improving the response to checkpoint inhibitor treatments [29]”.

The main goal of this study is to develop antibodies that could be used to vectorize agents acting on DNA damage repair mechanisms like PARPi. The authors did not explain this in their study in a clearer manner. The rationale is not clearly defined.

Response: This is an excellent observation. We acknowledge this and we have reformulated the article in accordance, highlighting the concept that we are identifying targets for vectorized biologics in different formats, including peptides, and not only antibodies.

Round 2

Reviewer 3 Report

Comments and Suggestions for Authors

The authors have removed most of the description of the statistical analysis, leaving only references to the construction of figures and charts. However, Figure 5 is not a figure but a table presenting the results of the correlation analysis, while Figure 4b presents the results of a regression analysis, which I assume to be the Cox regression. These analyses should be described in the Statistical Analysis subsection. Additionally, these figures would be better presented as tables rather than figures.

Author Response

Comments and Suggestions for Authors

The authors have removed most of the description of the statistical analysis, leaving only references to the construction of figures and charts. However, Figure 5 is not a figure but a table presenting the results of the correlation analysis, while Figure 4b presents the results of a regression analysis, which I assume to be the Cox regression. These analyses should be described in the Statistical Analysis subsection. Additionally, these figures would be better presented as tables rather than figures.

Response: 

We thank the reviewer for their helpful feedback and the opportunity to provide clarification. We would like to emphasize that the descriptions of the statistical analyses have not been removed. In fact, to ensure transparency and precision, we have added several clarifications in the revised manuscript to explicitly state the source and type of each statistical test used, including:

  • Line 174: “assessed by the Mann–Whitney U test”

  • Line 202: “Cox proportional hazards regression analysis was made for each gene separately”

  • Line 241: “Correlation data (purity-adjusted Spearman's rho)”

Additionally, we have provided corresponding explanations in the Materials and Methods section to describe each statistical approach applied throughout the study. To further enhance clarity, we have also introduced a dedicated Statistical Analysis subsection within the Materials and Methods (lines 134 - 142 in the revised manuscript).

With regard to the figures referenced:

  • Figure 4b presents the results of Cox proportional hazards regression analysis, generated via the KM Plotter tool. It is certainly a table, but we believe that it is not very explanatory on its own and that it makes more sense together with Figure a and Figure c, so it is part of a figure panel rather than a separate table.

  • Figure 5 is not a table, but a heatmap visualizing Spearman’s correlation coefficients and associated p-values, derived from TIMER2.0. The heatmap format was selected to facilitate comparative visualization of gene–immune cell correlations across multiple datasets.

We have updated the manuscript accordingly to reflect these clarifications.

Reviewer 5 Report

Comments and Suggestions for Authors

The manuscript by Adrian et al. examined candidate genes expressed on the membrane surface that may serve as targets for therapeutic intervention in cancer.

The authors have addressed all the previous comments. Thus, the manuscript can be accepted in its present form.

Author Response

Comments and Suggestions for Authors

The manuscript by Adrian et al. examined candidate genes expressed on the membrane surface that may serve as targets for therapeutic intervention in cancer.

The authors have addressed all the previous comments. Thus, the manuscript can be accepted in its present form.

Response: Thank you for your previous comments, which have enabled us to improve, and for considering our work acceptable for publication.